# The Best Practice of CRM Implementation for Small- and Medium-Sized Enterprises

**Michal Pohludka [1] and Hana Štverková [2,*]** 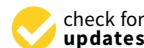

1   ZKV Career, s.r.o., Chomutovska 38/10, 161 00 Prague, Czech Republic; michal@zkvcareer.com
2   Department of Business Administration, Faculty of Economics, VSB—Technical University of Ostrava, 17. listopadu 15/2172, 708 33 Ostrava, Czech Republic
*   Correspondence: hana.stverkova@vsb.cz

**Abstract:** The biggest key aspect to the success of a business is a satisfied customer. For this reason, it is possible to state that the growing trend of focusing on the customer and his/her needs has prevailed in recent years. The aim of this article is to analyze the use of CRM (Customer Relationship Management) systems in small- and medium-sized enterprises (SMEs) in the Czech Republic and to find the determinants for CRM system implementation. The best practice for CRM implementation suitable for SMEs is clarified using a specific case of a global enterprise. A fully functional CRM system can be considered a competitive advantage, and this is not only the case for global companies, but also for small and medium enterprises. Using a functional CRM interconnected with an ERP system, enterprises are able to manage business and direct marketing activities, as well as the company's overall profits. These functional systems lead to an integrated system called funnel management, which improves customer relationship management and leads to a sustainable business.

**Keywords:** CRM; customer; funnel management; key account management; needs; segmentation; small- and medium-sized enterprises; sustainable business

## 1. Introduction

Market trends have given rise to digitalization, the use of technology, and the transformation of businesses, in an era called Industry 4.0. The most key aspect to the success of a business is a satisfied customer. The business environment is changing year by year, and marketing and business practices have witnessed an ever-growing tendency to focus on the customer and his/her needs (Barreto et al. 2018), (Erkmen 2018) or (Brodie and Brush 2008). The focus on the product and the solution offered has been sidelined. That is why knowledge of customers, their focus, their needs, their financial possibilities, and their potential are crucial (Wu et al. 2018). Every customer is important. However, despite this fact, it is commercially necessary to classify customers based on certain criteria. The traditional goal of marketing strategies is achieved through the growth of new customers; customer loyalty becomes the main goal of customer relationship management. It is usually more beneficial and less costly to keep existing customers who shop regularly and in large volumes than to constantly try to find new customers (Kotler and Keller 2013). The companies on the market are aware of this trend, which leads to their introduction of various CRM (customer relationship management) systems. For global companies, priority is given to those with a global reach, predominantly monolingual and "cloud".

At a fundamental level, global companies operate in a very similar way, regardless of their segment of activity. They have an appropriately arranged organizational structure (Stverkova and Pohludka 2018) and strive to provide employees with all business standards and complete business operations in the form of the CRM system, reporting and calculating the

implementation of individual aspects of the business. A system set and operating in this way is an essential prerequisite for business management, customer segmentation, and subsequent effective marketing activities and sustainable business, even for small- and medium-sized enterprises.

In recent years, there has been an increased focus on the CRM system from the software point of view and the necessities for the profitable relationship. Findings from various studies have justified the need of CRM for long-term business sustainability (Siu 2016). Prior studies have generally found a positive relationship between implementation of the CRM system and company performance (Arsić et al. 2019), or (Siu 2016).

The current research gap, also considered the missing piece in the research literature, is the area of determinants of the CRM implementation process and benefits, which have not been explored enough. With a change in the size of a company, there arises the need to implement the CRM system. Together with a change in entrepreneurs' attitudes and perceptions regarding the business sustainability and the consequent changes in the purchase behavior of the consumers, there arises a need to study and analyze the economic, socio-cultural, and psychological factors affecting the decision to implement the CRM system in small- and medium-sized enterprises. Therefore, the aim of the article is to analyze the use of CRM systems in small- and medium-sized enterprises in the Czech Republic and to find the determinants for CRM system implementation. The best practice for CRM implementation suitable for SMEs will be clarified using a specific case of a global enterprise. Related to previous knowledge of SMEs, implementation of the CRM system in global companies, and the trend of digitalization, the authors need to address the topics by defining the following research questions:

RQ 1: "What are the key steps for SMEs, in terms of the determinant of CRM system implementation based on the best practice of global companies?"

RQ 2: "What can transform SMEs from the CRM system perspective into a globally-owned enterprise?"

The answers to these questions lie within a systematic literature review, and demand conducting empirical research to fully examine dependencies between processes in the SMEs and important determinants of the CRM system implementation. It can be claimed that key customer relationships in SMES have not been clearly analyzed (sufficiently) in practice, and rarely in theory. Ascertaining the level of involvement of the owner is essential when analyzing key customer relationships, with respect to specific business-to-business and business-to-consumer relationships. Furthermore, there has been a lack of quantitative data related to successful implementation of the CRM system for SMEs, and an even greater lack of deep, integrated research about this topic.

One impact of globalization is the constant growth in competition. As the availability of goods expands, the independence on product localization increases, and marketing has an ever-increasing impact at a global level. In this environment, companies struggle to find a competitive advantage and sustainable development (Malik and Jasińska-Biliczak 2018). All companies' business activities, either multinational corporations or small- and medium-sized enterprises, SMEs, are affected by sustainability issues. Although SMEs are smaller, collectively, they have a significant environmental and social impact. Thus, SMEs should adopt more sustainable behaviors and sustainable processes (Shankar et al. 2017).

One of the reasons for the CRM concept is the crisis of the classic marketing mix itself. The company must combine elements of the marketing mix to achieve its goals in individual markets. The goal is to maximize profits, turnover, customer numbers, etc. This article's attention is focused on the small- and medium-sized enterprises in terms of the utilization and implementation of CRM systems in the Czech Republic. Based on the empirical research, the current state of SMEs in terms of CRM systems use will be assessed, and the determinants on which small- and medium-sized companies will decide on the introduction of CRM will be determined. Thus, the next purpose of this article is to provide a description of the best practice of the CRM system implementation process in a corporate company with a follow-up customer analysis and segmentation, including the creation

of key account management. This process can be considered as a follow-up activity after creating an appropriate business structure that suits the market and market trends according to Accenture Technology Vision, also for small and medium enterprises (Accenture Technology Vision 2017).

One of the main requirements for a CRM system is its availability from anywhere in the world. Gone are the days when CRM systems solely worked on customer databases, business management, and activities within it. Nowadays, they are interconnected with the ERP system of the company (Pohludka et al. 2018). Furthermore, details about the supply of goods and complaints can be seen in one place. The manager can see financial documents at any time, such as individual invoices. Another absolutely crucial feature of the CRM system is the capacity to approve individual transactions depending on the approval matrix and to thereby directly control the level of a company's profits. Such solutions also have their own pitfalls. They are robust, but not very flexible and require the right set-up of the entire system, including rights for individuals. Thus, thanks to funnel management, the manager is able to fully manage all his/her activities with the ability to report and manage business in order to see only his/her team (Horvathova and Davidova 2014).

A fully functional CRM system becomes the competitive advantage for global companies. Within a single system, it is able to manage the business and direct marketing activities, but it is also able to manage the overall profits of the company and has interconnected operations between business, logistics, technical, and service components of the company. This functional system leads to an integrated management system called funnel management.

The article is divided into three sections. First is the theoretical background focused on SMEs and CRM systems and methodology. The second part contains empirical research focused on the use of the CRM system in SMEs in the Czech Republic. The last part includes the CRM system implementation process in a global company as the best practice guide for use in SMEs.

## 2. Materials and Methods

### 2.1. Customer Relationship Management

In general, there is no uniform CRM definition. Each author defines CRM in a different way. Customer relationship management is primarily about harmonizing customer strategies and business processes, all in order to increase customer loyalty and business profitability. The cooperation of the parties ensures very close personal, informational, and operational connections, along with the achievement of long-term common goals (Dyché 2002; Hutt and Speh 2010).

Sustainable and up-to-date customer segmentation can only be provided in a healthy company if the CRM system functions well. If CRMs contain current data and this data is of a detailed nature, including the interest of the product portfolio, business potential, etc., then this segmentation can be fully utilized for many purposes.

"CRM is an interactive process that aims to achieve an optimal balance between corporate investment and the satisfaction of customer needs. The optimal balance is determined by the maximum profit of both the parties" (Chlebovsky 2005) or (Lehtinen 2007), (Storbacka and Lehtinen 2002) or (Siu 2016). A prerequisite for achieving this optimal state is to establish long-term partnerships with customers. Long-term perspective cooperation brings significant value to both parties that is expressible in monetary value.

According to Barreto et al. (2018), Chlebovsky (2005), or many others, customer care includes the continuous updating of customer needs, motivation, and habits; quantification of the benefits of key CRM functions; marketing, sales, and service activities; utilization of customer knowledge and experience in innovation of products offered; integration of marketing, sales, and customer support in a single whole; the use of modern tools to support customer needs; and quantification of the benefits of CRM, maintaining a balance between marketing, sales, and service activities in order to maximize profits.

CRM is a system that tracks customer interactions with the company and enables employees to find the necessary customer information, such as past orders, service history, problems solved, etc. (Lehtinen 2007; Storbacka and Lehtinen 2002). All records are conducted and used for the sole purpose of making the customer happy because it is the customer who makes the business run (Nguyen et al. 2007).

### 2.2. Setting up a CRM System

Setting up a CRM system primarily consists of creating a database of all customers, both current and potential. As much information as possible, in addition to the basic information, must be filled out, depending on the segment, the company's product structure, and the knowledge of the individual customers. This information is crucial to all marketing activities and to customer segmentation.

The term CRM is usually understood as corporate philosophy or, alternatively, a strategy that focuses on reducing costs and increasing the company's profitability by building long-term relationships with its customers. It is a summary of information processes and technologies (Technology Advice 2018), which brings benefits to the company in the form of loyal customers (Kumar and Shah 2004) and better use of cross-selling activities to help build the company's good reputation.

Therefore, CRM is perceived as a data-related technology that allows for higher profits or a strategic approach that brings value (Triznova et al. 2015). The advantage of CRM may be that it improves customer service, reduces costs, and better maintains clients (Vaish et al. 2016), (Kim and Kim 2009). The main goal of CRM is not profit, but the creation of value. Value should be created on both sides: both on the company's side and on the customer's side (Triznova et al. 2015).

CRM consists of three major elements, among which there is an immediate link, and the fourth element complements them. The four elements are (Wessling 2003):

- People—human capital, customers, employees;
- business processes—orientation, blending, unification;
- technologies—type, scope, area of use;
- contents—data, processing, sorting, archiving.

### 2.3. Customer Segmentation

Customer segmentation means dividing the market into separate customer groups that share similar characteristics. Segmenting customers is possible from many different perspectives. There are two basic ones: segmentation by product focus and by turnover or, as the case may be, potential (Kim and Kim 2009).

Product segmentation enables you to have a powerful way to identify customer needs, or even create needs. It allows for targeted and narrowly segmented marketing, which is much more effective than a global approach. By linking sales analyses and marketing segments, companies are also able to identify untapped market segments and quickly achieve leadership and thus a competitive advantage. Specifically, some products are good sellers, so this approach is applied to the entire segment of customers with the same focus. Global companies make use of this across different countries and continents, where portability is very fast thanks to a functional CRM system. This competitive advantage of global companies is widely used primarily in relation to local players.

Product segmentation also has a significant impact on company marketing. Therefore, the CRM system is one of its major platforms. It also allows a significant number of marketing functions:

- The quick launch of newly developed products/knowledge of the target group;
- Development of customer marketing programs/targeted marketing;
- Selection of a specific product portfolio for a given customer segment;
- Design of optimal distribution strategies or entire business structure targeting different customer segments.

In addition to a proper setup, daily maintenance and updates are essential for the proper use of the CRM system for marketing activities. Only when these conditions are met can it be used in the long term. When implementing CRM in existing organizational structures (Stverkova and Pohludka 2018) or (Tsasis et al. 2013), it is necessary to devote some time to improving staff qualifications, technological equipment, business process orientation, and data management. Introducing CRM is only possible when merging individual elements into a single whole. The one-sided view of CRM condemns the approach to failure in advance. A number of efforts to implement CRM did not achieve success because emphasis was only put on one of its elements (Wessling 2003).

The second segmentation of customers, which is used by almost every company, is segmentation based on turnover and purchasing potential (Birkinshaw et al. 2001). Each company approaches this segmentation in different ways. However, they have a common focus on key customers with high turnover or high potential. This focus is called "key account management" (Leischnig et al. 2018) or (Guenzi and Storbacka 2016). The basic idea is that it is necessary to take more care of customers who deliver the critical part of the turnover than others. In many companies, significant customers are assigned to their private sales manager (Key Account Manager), whose task is to map and meet their needs, develop personal relationships with key individuals, and build barriers against competitors. In addition to turnover, another criterion for the customer's inclusion in the key customer group (KA) is his/her purchasing potential for the future and the ability to influence other customers in their purchase decisions ("Key Opinion Leaders").

Introduction of key account management and customer segmentation based on the size of turnover and purchasing potential has the following sequence of events:

- Determination of which customers are crucial for a given company and segmentation into three groups, e.g., ABC is carried out (Balzan and Dall'Agnol 2017). For this purpose, various analyses and rules are used. The most frequently used ones are RFM—Recency, Frequency and Monetary; CLV—Customer Lifetime Value (Fader et al. 2005); and of course, long-term purchase history of the individual customers (Kumar et al. 2018);

- In cases where a company assigns a person responsible for key customers, it is important for the company to set sales and other goals, competencies, and cooperation with other team members (Pohludka 2018). If this position in the sales team is missing, then it is executed in most cases by cooperation between the sales team and the management of the company (Brakus et al. 2009) or (Edmondson and Harvey 2018);

- After introducing the classification of ABC customers and creating these three segments, it is crucial for marketing purposes to fill in all the available information of the product focus (Hanzelkova et al. 2013) or (Hutt and Speh 2010);

- For each customer, his/her map of competences should be set including organizational structure and decision-making authority. This is important for every business negotiation, and a map of competencies is frequently used in various sale and sale-strategic methods, for example, Blue Sheet (Strategic Selling Approach—a reference will be provided if necessary) more in Pollanen et al. (2016) or (Mintzberg 1994) or (Ajzen 1991);

- Competitive analysis and SWOT analysis should be carried out for each of them to find competitive advantages and potential threats. Based on these analyses, the company's marketing and business strategy is set and this can vary for defined customer segments (Farah and Gomez-Ramos 2014);

- Action plans with defined targets and strategies need to be prepared. Individual check points are set when success/failure is recapitulated Mintzberg (1994) or Hanzelkova et al. (2013) or Pohludka (2018).

Well-created customer segmentation provides companies with great business, as well as marketing, potential. When combined with a functional CRM system that is also linked to the company's ERP system (Pohludka et al. 2018), it means a strong competitive advantage that can be

used quite easily. However, the influence of the management of the whole company and its ability to motivate people and effectively manage and use the system remains considerable. Only after the team grasps the whole problematics correctly across all of the company's components can this potential then be capitalized and transformed into increased sales and market shares.

### 2.4. Small- and Medium-Sized Enterprises

The small- and medium-sized enterprises are usually presented as hidden giants. The SMEs in the European Union represent 98.84% of the total number of enterprises and 99.85% in the Czech Republic (European Commision 2018). They have a key position in the national economy in terms of creating healthy entrepreneurial surroundings and are considered as a main factor of economic development. Currently, in the European Union, SMEs in the non-financial business sector account for almost 67% of manpower employed and create 58% of value added generated by the non-financial business sector (European Commision 2018).

The importance of the existence of SMEs also illustrates the fact that the sector of SMEs in the Czech Republic shares more than 56% of the performance and added value and 60% ensures jobs with low capital costs. The medium-sized enterprises have been able to create 15% of new jobs in times of high unemployment and the small enterprises have even resulted in up 40% of new jobs (European Commision 2018; Stverkova 2013).

The official definition of SMEs takes account of three different factors: the level of employment, level of turnover, and size of the balance sheet (see Table 1). For all mentioned categories of enterprises, the independence has been another necessary condition, i.e., no other person or more people together, who fulfill the above mentioned requirements, cannot have a 25% or larger share in the fixed capital of an enterprise or its voting rights (Official Journal of the European Union 2018).

**Table 1.** Definition of SMEs.

| Company Category | Employees | Turnover | Balance Sheet Total |
|---|---|---|---|
| micro | <10 | <€2 million | <€2 million |
| small | <50 | <€10 million | <€10 million |
| Medium-sized | <250 | <€50 million | <€43 million |

Source: Commission Recommendation of 6 May 2003 concerning the definition of micro-, small-, and medium-sized enterprises (2003/361/EC), Official Journal of the European Union, L 124/36, 20 May 2003 (Official Journal of the European Union 2018).

Except for the above-mentioned classification of small- and medium-sized entrepreneurship, which represents the quantitative classification, the qualitative aspect of SMEs also exists. The qualitative determination of SMEs has been an important prerequisite for a correct understanding and definition of this group (find more in Stverkova (2013)).

If the company wants to be competitive, it must be perfectly oriented in the globalizing entrepreneurial surroundings. The essence of competitiveness is thus a conscious creation and maintenance of competitive advantages. It is necessary that companies, instead of eliminating disadvantages, search for ways to grow and build the advantages. One option is to implement the customer relationship management system. One of the most primary research requirements is to obtain information to help predict the key factors for choosing the correct CRM system.

### 2.5. Research Framework

In order to meet the aim of the article—to analyze the use of CRM systems in small- and medium-sized enterprises in the Czech Republic and to find the determinants for CRM system implementation and solve the research questions, in terms of utilization and implementation of CRM systems in the SMEs in the Czech Republic—we conducted a marketing survey in selected small- and medium-sized companies in the Czech Republic, which was realized in the period of March to June 2018. In the first phase of the research, a method of selecting a suitable sample, sample size, appropriate

methods, and survey tools was established. The respondents consisted of economic entities classified according to the Commission Recommendation of 6th of May 2003 concerning the definition of micro-, small-, and medium-sized enterprises (2003/361/EC). The size of the research sample is determined on the basis of a formula according to Nowak (Chraska 2007). The empirical research was carried out by means of desk research and a questionnaire survey completed in SMEs, and was supplemented by secondary research on the determinant for CRM system implementation. The first questionnaire was divided into two parts. The first part of the questionnaire survey focuses on defining the size of the business in terms of the Commission Recommendation definition of SMEs, and the second part is on the current state of SMEs in terms of how CRM systems are assessed and the factors on which small- and medium-sized companies make the decision to introduce CRM. The second aspect of the research focuses on the determinant for CRM system implementation.

RQ 1: "What are the key steps for SMEs, in terms of the determinant of CRM system implementation based on the best practice of global companies?" This research question is solved on the basis of best practice in a global company. The purpose of this article is a description of the process of deploying a CRM system in a corporate company with a follow-up customer analysis and segmentation, including the creation of key account management. This goal is described in a case study in a global enterprise. The case study approach is one of the frequently used ways of conducting social science research (Pohludka et al. 2018).

Every method has peculiar advantages and disadvantages, depending on the research questions, actual behavioral events, and focus on contemporary development. According to Yin (2002), the case study has been a common research strategy in business, social work, planning, and sociology because it allows an investigation to retain holistic and relevant characteristics of real-life events. Hence, it can be said that case studies are increasingly used as a research tool to investigate a research issue. To conclude, a case study should be easy to understand, in order to fulfil its purpose (Pohludka et al. 2018) or (Meyer 2001) or (Hickson et al. 2003).

## 3. Use of the CRM System in SMEs in the Czech Republic

Customer relationship management is one of the important aspects of competitiveness in the 21st century. All companies, regardless of size and focus, must concentrate on the customers, and their satisfaction and loyalty. In small- and medium-sized businesses in the Czech Republic, the survey of the current state of CRM utilization in small- and medium-sized enterprises was conducted in the months of March to June 2018. A questionnaire survey was used. In total, 860 enterprises were approached, and the return on the questionnaire survey was 37%, i.e., 319 enterprises from the sphere of small- and medium-sized enterprises provided feedback. The aim of the survey was to determine the use of a CRM in small- and medium-sized enterprises and the factors which small- and medium-sized companies consider to decide on the introduction of the CRM, why the CRM has not yet implemented the results of companies, and to define appropriate recommendations. Among the small- and medium-sized enterprises were business units according to EU typology, according to the EU regulation (see the Table 1).

In the small businesses, it is necessary to set expectations for the decision to implement the CRM. In the field of small- and medium-sized businesses, the most important CRM decision-making criteria are whether efficiency results are achieved, as businesses also expect to work better with their customers, manage their customers, and increase their value. A no less important factor was, according to quantitative research, that they expect to strengthen their own brand. The very purpose of implementing the CRM in an enterprise is to increase revenue, customer satisfaction, and the number of new customers.

It was found out that out of the 319 surveyed entities, only 58% of enterprises use the centralized information system. In total, 81% of respondents collect data about their customers, but this does not mean that a CRM is implemented in the company. The CRM philosophy is a prerequisite for an enterprise to collect data on its customers and work with them. As registered components, companies

can use, for example, programs from the Microsoft Office suite, and paper records have been recorded for small companies. The first chart (Figure 1) shows the use of the centralized information system and customer data collection in the investigated enterprises by their size. It is perceptible that medium-sized enterprises use the centralized system more than micro-enterprises. It is possible to state, based on collecting data and using centralized IS in SMEs, that readiness to implement the CRM system is weak.

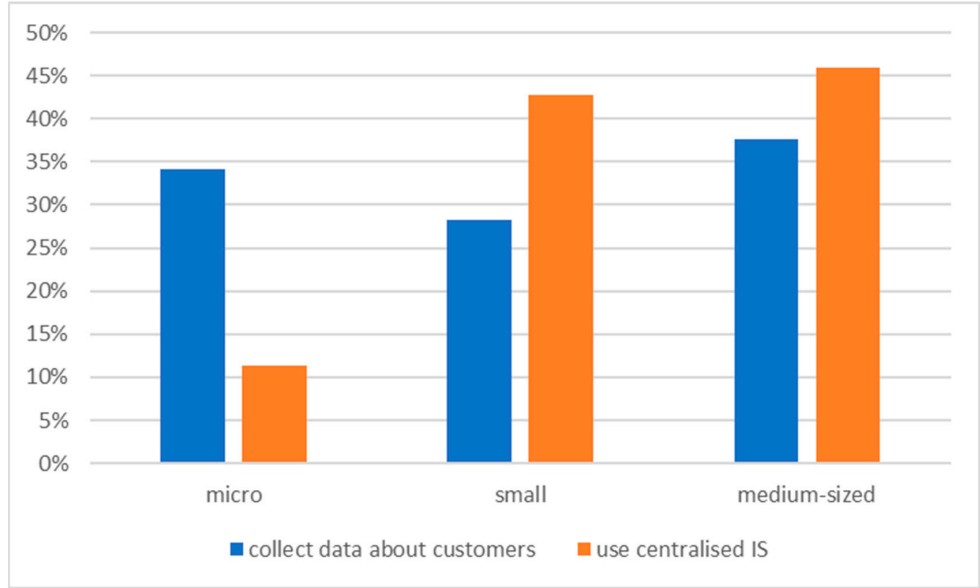

**Figure 1.** Readiness to implement the CRM system.

In terms of the communication with customers, the most frequent form was electronic, including 93% of respondents, 78% by mobile phones, and 56% of personal contact with the customer, which appear to be key to small and medium enterprises. Nowadays, modern ways of communication are through modern social networks (media) (Instagram, Facebook . . . ) and messengers—instant messenger, WhatsApp, Viber, WeChat, etc. (21%). The type of communication with customers in small- and medium-sized enterprises is shown on this chart (Figure 2). It is visible that micro-companies use direct marketing and it is typical to use social networks for communication, which in their opinion, is the easiest way to communicate with customers.

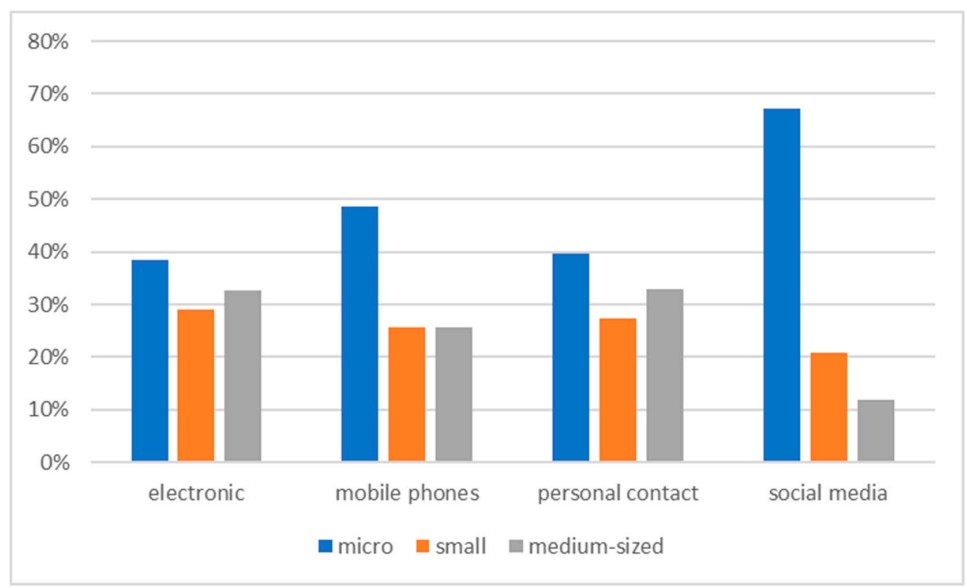

**Figure 2.** Type of communication with customers in SMEs.

Only 68% of the SMEs surveyed are still working on the acquired customer data as they see an improvement in customer relationships (53%).

The key issue of the whole research is to verify the implementation of CRMs in the companies surveyed. Of all 319 entities, 45% have a CRM system that is in place; 3% are in the implementation phase; 21% are considering deployment; and 31% of the respondents are not considering CRM implementation, especially micro-enterprises, which do not think about it. For medium-sized businesses, CRM has been introduced in 80% of surveyed entities. The implementation time for these enterprises has been, in most cases, one year. The research results are shown in Figure 3.

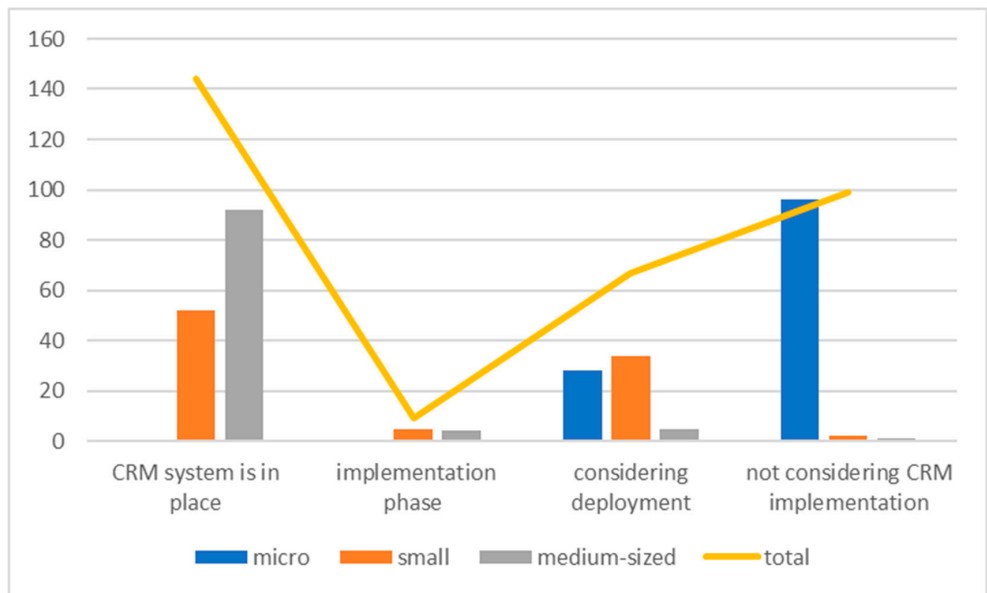

**Figure 3.** The CRM implementation in SMEs in Czech Republic.

The reasons why SMEs did not implement CRMs is based on insufficient information, insufficient technical background, and the cost-effectiveness of CRMs for implementation and use that these companies consider to be superior to their business.

The reasons that appear to be key to SMEs for deploying CRM are:

- increasing efficiency,
- customer satisfaction,
- customer loyalty,
- brand reinforcement,
- costs reduction.

The criteria for choosing the right CRM can be analyzed using a multi-criteria decision, Saaty's method, Fuller's triangle, brainstorming, and others. Based on the results of the questionnaire survey, the following five key criteria were identified as being the most taken into the account when making the decision about CRM.

The authors conducted a survey of 621 respondents, including representatives of global businesses, on CRM implementation criteria. The most significant criteria were perceived to be CRM reliability (81%), user friendliness (68%), support and service (58%), compatibility with other ISs in the enterprise (37%), and price (35%). Detailed results of the survey are shown in Figure 4.

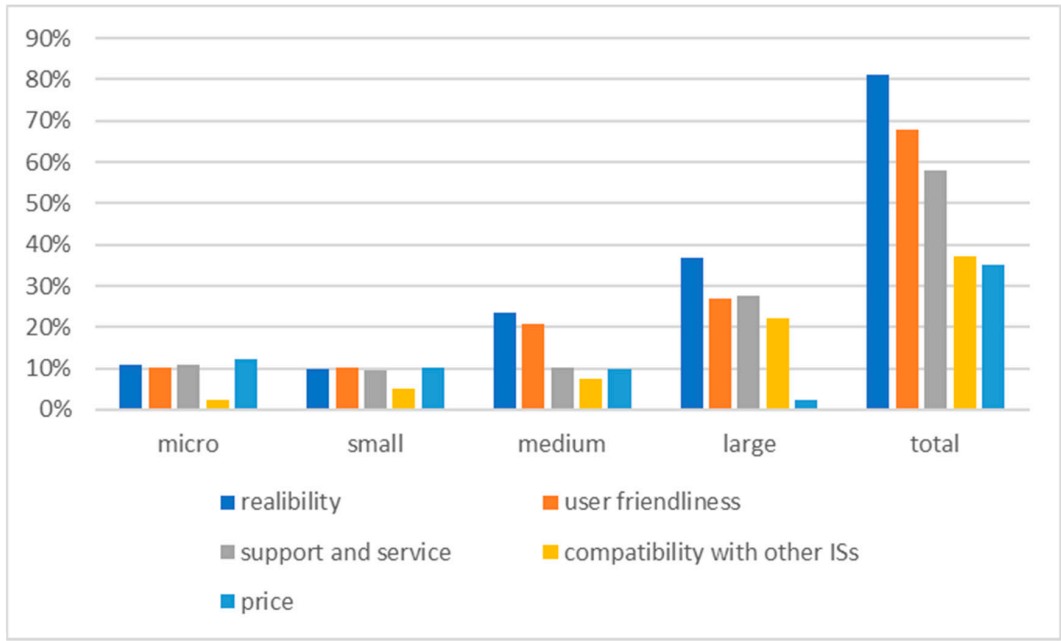

**Figure 4.** The CRM implementation criteria by respondents.

When evaluating the SMEs using CRMs, there are especially companies operating in the IT, logistics, construction, and services areas.

Surveys show that most small- and medium-sized businesses in the Czech Republic have not yet implemented CRMs, especially due to a lack of awareness. Small- and medium-sized enterprises have their own specificities that need to be worked on in determining the right solution. These specifics include narrow specialization, limited access to capital, a flat organizational structure, and direct customer contact. However, the criteria for CRM deployment must be set individually by each SME, depending on its size, financial resources, and industry. For every business, regardless of its size, it is considered together with the achievement of business goals and the goal of achieving profits. In order to be successful in this regard, the customer is the main prerequisite, including his/her satisfaction and his/her loyalty. It is therefore logical that the main benefit that companies expect from CRM is also an overview of their customers, facilitating their work with information about them, and the goal is to more efficiently achieve expected business results, resulting in more sales, customer satisfaction, customer sales growth, and increasing turnover.

## 4. Introducing the CRM System with a Follow-Up Customer Segmentation

A global company set the goal of introducing a CRM system worldwide. This is a system that will be the same in all countries of the world, and that will be intuitive and timeless, and, from the point of view of business management, will allow "funnel management".

The company management decided on a global solution and one supplier of a CRM system, which is "salesforce" (www.salesforce.com). This system was implemented in each country where the company has direct representation and the license was provided to all the employees from both the sales and marketing departments.

In this part, the CRM system implementation process will be described, and this should be the best practice guide for SMEs. It includes RQ 1: "What are the key steps for SMEs, in terms of the determinant of CRM system implementation based on the best practice of global companies?"—See Figure 5.

When choosing a CRM system, it is necessary to define the needs and requirements and choose a supplier based on these. One of the key aspects of the optimal choice is the selection from established suppliers and proven brands. The current situation on the market with CRM systems shows that the following systems are among the leaders:

- Salesforce—focused on clarity and tries to simplify work of the user as much as possible. There are shortcomings in the area of analytics and e-commerce;
- SAP CRM—obtaining instant overview of the customer and his/her needs. The main shortcomings are high cost and long implementation;
- Oracle Siebel CRM—is available in many various modifications tailored to the specific needs of specific industries. The main shortcomings are the high cost, long implementation, and the necessity of integration with other systems;
- Microsoft Dynamics 365—most demanding in terms of implementation. Along with the system, it is necessary to own or install and use other Microsoft products;
- Pegasystems—costly;
- SAP Cloud for Customer—weaker in marketing and in customer care;
- Helios Orange—targets small- and medium-sized companies.

The selection of a suitable CRM system can be done using the multi-criteria evaluation of the variant, by Saty's method or Fuller's triangle. Implementation of the CRM system was carried out in several stages. The first stage was the comprehensive training of the employees in the question of how to use the system. The second stage included interconnection with the ERP system, in the case of this company, the SAP system, with salesforce.com so that customer names (institutions, thus legal persons) and their reference numbers were identical. Thus, their interconnection was assured, as was subsequent continuity between business management and financial parameters, such as sale results and statistics that are obtained from the SAP system. Within the last stage, the customers as such, who work in the given institutions, were entered into the system. Basic and contact information was assigned to every person. It was crucial to add the area of interest for follow-up marketing activities to each person.

The training of all employees as such in the EMEA zone (Europe, Middle East, Africa) and then in individual countries took four weeks. It consisted of three stages. In the first stage, it was provided to managers through in-person weekly training. The reason for this was, in addition to describing the system, to explain the reason for its use and the overall vision of the company. Through this training, responsibility was given to middle management to introduce the CRM system in individual countries. In the second stage, teams were trained in each country where the CRM was planned to be used and in the last stage, just before initiation of the real operation, there was online training and a discussion about the unclear functions and expectations of the company's management.

The onset of use was slow. From the point of view of people's management, no pressure has been placed on the employees to avoid their demotivation, or even the increased outflow of people. After two months of using it purely for the purposes of time management and business dealings, people began to use the CRM system intuitively, and it was possible to move into the next stage, i.e., segmenting customers and introducing key account management.

Segmentation of customers had already been carried out in the CRM system, and the customers were classified into three groups according to the ABC model. According to the actual companies discussed, the customers were classified as "gold, silver, and bronze":

- Gold customers constitute a maximum of 10% of the total number of employees. In terms of turnover and business potential, these are the biggest customers. According to sale statistics, these 10% of customers constitute 60% of the total turnover of the company. Moreover, they have, apart from a high turnover, an additional positive effect. They influence other, less significant customers to buy the goods of the company through references and they act as Key Opinion Leaders;
- Silver customers constitute approximately 25% of the total number of employees. They have a smaller turnover, but they have the potential of percentage growth, not in absolute monetary values. This set of customers is important, especially for its growth potential. It is a set of customers who, according to the sale statistics, represent approximately 20% of the total turnover. However, given their potential, they are possible drivers of growth;

- Bronze customers are all the remaining customers who do not have large or average turnovers, and the growth potential is limited. These customers are only rarely visited by sales representatives in comparison with silver and gold customers, and the marketing concept sales approach is used here instead of personal negotiations.

With regards to customers of the "Gold" type, these are the key customers for the companies. The first approach of the company is usually assigning a separate person who is directly responsible for selling to these customers. In the company described, they chose a different approach; the reason for which was a lack of workforce. The nine most significant customers were chosen from the Gold group out of the total number of twenty customers, and these were considered the key customers for whom the key account management program was developed. These customers constituted 40% of the total turnover. The program consisted of regular visits not only by a sales representative, but also by a sales director. For each of them, a program of expert educative activities was created, which is, in the area of health care, a substantial competitive advantage, provided that the company is able to provide the expertise. Last but not least, these customers were offered special terms of both price and loyalty, with the aim of building long-term cooperation. After one year of intensive work on this project, positive results were obvious, when sales within these key customers increased significantly (year-on-year ten percent). As part of a satisfaction survey (NPS Score) of customers, two different profiles for "gold" customers and the remaining ones were obtained, where the feedback from the significant customers is much more positive (Hickson et al. 2003; Hong and Kim 2002).

After successfully introducing a CRM system, a system of managing business opportunities and funnel management was implemented in parallel, along with customer segmentation and key account management. The basic philosophy of funnel management is the classification of individual business opportunities into several groups, according to the stage of business process:

- Lead—Opportunity as a result of a marketing campaign or created by a colleague (support or customer service etc.);
- Opportunity Identified—A customer situation or enquiry that suggests dissatisfaction with an existing product/service or that a potential need exists. Budget/timeline/exact product or service typically not yet known;
- Opportunity Qualified—The timeframe, budget, and general requirements of the customer are clearly understood and it is possible for Bio-Rad to satisfy these requirements;
- Negotiated—Customer accepted our solution from a technical perspective. Objections clarified and drawbacks outweighed. Customer and Sales team are now working on commercial terms and conditions;
- Accepted—Customer has now verbally accepted our solution. Waiting for the official purchase order to arrive. The close date represents the expected arrival date;
- Close Won—Purchase order received and/or contract signed to start providing products or services.

The goal is to make every single business opportunity "move" through individual stages to a successfully finalized deal. When properly used, a real set of business opportunities is created at different stages of the business process, which should serve as the source of all business opportunities on which it is necessary to make effort, according to importance and topicality.

Currently, the CRM system already allows much more. A very valuable feature is contract management, when a contract is created from each won business opportunity subsequently saved in the CRM system. In the case of business conditions where there is a continuous takeover of goods on a regular basis, this contract can also be divided from the time viewpoint and included in the forecasting activities. This module was only implemented in the last stage and falls under the remit of the Sales Support Specialist because, in addition to entering data into the CRM system, the contract must also be physically stored according to ISO 9001 standards.

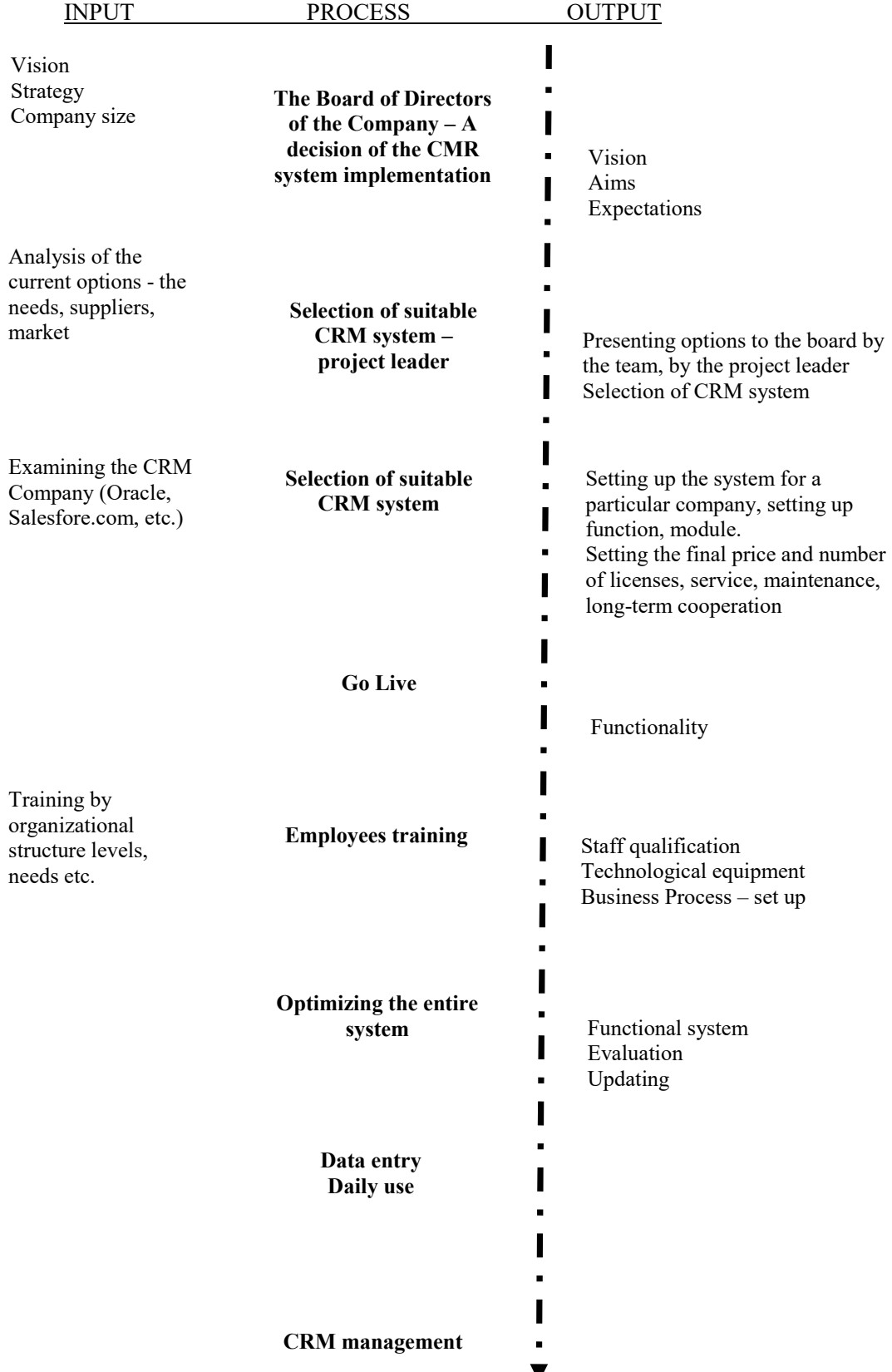

**Figure 5.** Key steps for SMEs, in terms of the determinant of CRM system implementation based on the best practice of global companies.

## 5. Conclusions

After more than two years of the routine use of salesforce.com's CRM system, the system contains more than 1500 contacts with all relevant data for sales and marketing activities. Such a robust system has become the cornerstone of business management, launching new products and marketing events. Key account management or taking care of key customers is important in all businesses in both the consumer and industrial markets.

All of the company's operations have been interconnected in terms of software. Nowadays, the ERP system is interconnected with the CRM system in the company. These two core business systems were linked to all other ones related to the financial, marketing, and business agenda. This fact enables easy and straightforward tracking of sales statistics, current sales numbers, and sales forecasting. The homogeneity of the system also allows the central administration of ISO documentation and documentation for financial audits. Last but not least, these systems are automatically connected to the approval system of prices falling under discounts or transactions with the investment component of the company.

RQ 2: "What can transform SMEs from a CRM system perspective into a globally-owned enterprise?" The key to a successful CRM initiative is correct and consistent customer data accessible online across the whole IT infrastructure. It is important to realize that CRM solutions affect both sales and marketing, while maintaining customer satisfaction. Customer relationship management is a strategy that focuses on building and supporting long-lasting customer relationships. It is not just a technology, but a change in the philosophy of the company, so the emphasis is placed on the customer. Failure to comply with this strategy is usually the reason behind most of the failures to implement CRM systems more in Pohludka et al. (2018); Siu (2016).

Among the indicators of necessity of CRM are customer information chaos and an inability to assess customer value. For example, it may be argued that an indicator of the need to purchase a CRM system is the need to reduce the sales cycle, increase the number of key 14 performance indicators, or increase the productivity of service workers and loyal customers. If the company is not able to determine how many customers it has, what products the customer owns, how much he/she has lodged a complaint, or how his/her total transaction volume or turnover has changed over the last period, the CRM is very much needed (Wheatcroft 2018; Finnegan and Willcocks 2007).

From the point of view of feedback, it is necessary to have an established system for evaluating the functioning of the systems. It is clear that not many companies on the market are nowadays able to avoid customer relationship management if they want to increase or generate higher profits. It does not matter how large their market share is. With the use of information technology, the world market becomes more accessible, so companies need to be more competitive. Customers are well-aware of a wide range of products and their alternatives, so their main concern is often not only the price/quality ratio. Instead, relationships and product-related services are increasingly important. Companies are aware that if they understand the needs of their customers, it will mean returning customers for them and therefore relatively easily attainable recurring profit. It is necessary to distinguish between CRM as a strategy and CRM as a software.

Measuring the efficiency of CRM should combine long-term visions, strategies, and goals in CRM with specific short-term, tactical, action, and evaluation plans, which form the stimulus of CRM (Krizanova et al. 2018). The most widely-used methods that are suitable for measuring the efficiency of CRM are the balanced scorecard, Model CRACK, TQM, or Model EFQM.

In the current period, more intensive use of information technologies for collecting and storing information on clients can be observed. The further limitation of the efficiency of CRM stems from the fact that it is not possible to create a generalized CRM system for all companies in a particular sector, about SME see Chromcakova et al. (2017). However, the trend of using information technologies is also reflected in the use of innovative approaches to relationship marketing, leading to new types of customer relationship management. The way to attract and maintain loyal customers is to use management according to funnel management. Future research should address issues such as

identifying the importance of individual key areas in order to objectify the weights of each criterion in further empirical research.

**Author Contributions:** All the authors conceived and designed the research, analyzed the data, and wrote the paper.

**Funding:** This research received no external funding.

**Conflicts of Interest:** The authors declare no conflict of interest.

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
