# Peer review of "The Best Practice of CRM Implementation for Small- and Medium-Sized Enterprises"

_admsci, doi:10.3390/admsci9010022_

Round 1
Reviewer 1 Report
The paper needs revision. Introduction does not contain necessary issues: what is the input of paper, what is the main goal of the paper and how it is linked with scientific problem. The literature review in introduction does not provide any background. The structure of paper also not described in introduction. There is no theoretical methodology provided for the methodology applied in the paper. Discussion section is pour, conclusions also does not provided enough information on findings of this paper. Literature list also requires review by including more recent studies in this area.
Author Response
Honorable reviewer,
thank you for helpful comments. We have changed the title and the abstract of the paper.
The best practise of CRM implementation for small and medium-sized enterprises.
We defined aim and Research questions. As well as we introduced the structure of the paper. We add new literature. We have changed the aim and add the research results and recommendations for SMEs.
Reviewer 2 Report
The topic of the paper is relevant and timely, but the content is immature, aiming for a didactic or marketing goal rather than presenting a scientific research study. A survey and case study were applied, but they are underdeveloped, superficially discussed and buried very deep in a lot of background or generic information about the benefits of CRMs and customer segmentations.
The title is not a title, it is a full sentence (with predicate) and says nothing about the contribution of the study.
The abstract also does not say anything about the paper contribution. "The authors focus on the use of CRM systems..." is too vague. "The aim of this article is a description of the process..." - this cannot be a research goal! (it could be a didactic or journalistic goal). Is this a process proposed by authors or a process observed by authors (or proposed and then observed)? Otherwise it sounds like something that authors learned and try to share with us (which is not a scientific effort but rather a didactic one).
The introduction clarifies things a bit - however, it still mentions "a description of the process" without stating what is the relation between the study and that process, nor what the authors plan to do with that process and what should we learn from it.
The vagueness of the contribution is again obvious in section 2 - the first time that a research goal and method are mentioned is page 6 (half of the paper!). Even there, it is positioned as a "marketing survey" and not as a scientific empirical study. There's no detail about how this survey was built, measured - no statistical analysis is provided. There are no charts, figures, tables, only narrative reporting of some percentages. The paper could be pushed in the direction of Enterprise Architecture - showing how the introduced CRM changed some business processes, a process landscape or a service portfolio, but no attempts are made in the direction of a systematic model that could be useful outside the reported case.
A proper literature review section is missing therefore we have no positioning of the authors contribution against a body of knowledge baseline, we cannot grasp what this paper brings and was not already known. Section 2, which is an attempt of literature review provides only background information (of didactic value). A literature review should be filtered through a critical lens, establishing relations to the proposed contribution. Section 2 is not able to fulfil this role.
Other parts of the paper look like a rudimentary case study, not really developed along the lines of the Case Study research method. Authors should decide if they want to present a marketing survey, an empirical scientific study, an enterprise engineering case study, an ethnography etc. and tailor their work within adequate frameworks for the type of paper.
Scientific methods are briefly mentioned in vague terms "multi-criteria evaluation can be done..." A scientific journal is not interested in what can be done, but in what was done by the authors; and how it was done, towards which (nontrivial) results?
Authors should avoid stating the obvious. Sentences such as "Customer relationship management is one of the important aspects of competitiveness in the 21st century" do not inform the reader in any way. General statements and background information should be reduced to exactly those parts that are relevant to communicate a scientific study and to address a research goal that must be precisely formulated from the very beginning of the paper (abstract and introduction).
Author Response
Honorable reviewer,
thank you for helpful comments.
We have changed the title to The best practise of CRM implementation for small and medium-sized enterprises. and the abstract of the paper. We changed the introduction of the paper - aim, research questions. Add the more detailed research results with Charts, Figure and we tried to adjust the article according to the recommendations.
Round 2
Reviewer 1 Report
The paper is significantly improved. All comments and remarks by Reviewers were taken into account. Now introduction of paper is prepared according requirements of scientific manuscript. I recommend paper to be published in current form.